# A socio-ecological examination of the primary school playground: Primary school pupil and staff perceived barriers and facilitators to a physically active playground during break and lunch-times

**Michael Graham**[1]*, **Kevin Dixon**[2], **Liane B. Azevedo**[3], **Matthew D. Wright**[1], **Alison Innerd**[1]

**1** School of Health and Life Sciences, Teesside University, Middlesbrough, United Kingdom, **2** Sport, Exercise and Rehabilitation, Northumbria University, Newcastle, United Kingdom, **3** School of Human and Health Sciences, University of Huddersfield, Huddersfield, United Kingdom

* Michael.graham@tees.ac.uk

## Abstract

Using Brofenbrenner's socio-ecological model as a conceptual framework, the objective of this study was to determine playground users (primary school staff and pupils) perceptions of the barriers and facilitators to a physically active school playground at an intra-personal (individual), inter-personal (social), environmental and policy level. Results from a series of qualitative interactions with children (n = 65) from years five and six (9 to 11 years old), and structured interviews with adult teachers (n = 11) revealed key differences in the child and adult perceptions of the playground and the purpose of break-times. A number of inter-related environmental boundaries and school policies were identified as restrictive to children's explorations and activity levels during 'free play' periods, which centred on resource availability, accessibility and health and safety. Further, traditional playground hierarchies act to promote and prevent physical activity engagement for different groups (e.g. gender and age). Finally, differences between the adult and child perception of the primary school playground were observed. Playground physical activity, during break-times appears to be affected by a number of variables at each level of the socio-ecological model. This study provides an opportunity for primary schools to reflect on primary school playground strategies and practices that are implemented at each level of the socio-ecological model to encourage a more effective use of the playground during school break-times.

## 1. Introduction

High rates of physical inactivity have been reported among children of primary school age in the UK [1] and worldwide [2, 3]. Physical activity in this age group is important for a number reasons; such as improved cardio-metabolic health [4, 5], bone health [6], and mental health [7]. However, physical activity is a complex and multi-dimensional behaviour determined by numerous biological, psychological, sociocultural and environmental factors [8–12].

**Data Availability Statement:** All relevant data are within the paper and its Supporting Information files.

**Funding:** The author(s) received no specific funding for this work.

**Competing interests:** The authors have declared that no competing interests exist.

Ecological models of health (and physical activity) are one such method in considering a wide range of determinants. The socio-ecological model, originally developed by Brofenbrenner [13] and adapted by Sallis Bauman & Pratt [14], focusses attention on the key individual, interpersonal, environmental and policy agencies that have an active role in health and physical activity promotion.

There is evidence to suggest interventions will fail to make long term, sustainable changes to daily moderate to vigorous physical activity (MVPA) if they fail to adequately consider the interactive characteristics between individuals and their environment at the intra-personal (individual), inter-personal (social), environmental and organisational/policy level [15, 16]. For example, implementing changes at an individual level by encouraging engagement in physically active pursuits during break-times will only work if appropriate environmental and policy level changes are also implemented at the school. However, many childhood physical activity interventions do not consider the multi-level influences on children's behaviour during the intervention [17].

Despite the inconsistency associated with the design of physical activity interventions, it is universally accepted that interventions within the school environment are important, and for good reason [18]. Children between 5 and 11 years of age can spend up to 30 hours per week within the school environment [19] making it an ideal setting to promote physical activity. Within a school day, school break-times are reported to be the most favourable periods of the day for children [20], providing periods of time for children to *"catch up"* with their friends [21] which can positively impact on the integration and adjustment to the school environment [22]. However, as Baines and Blatchford [20] have indicated, schoolteachers (as part of the wider institution) and external policy makers tend to hold contrasting views. That is to say, school break-times are perceived by adult decision makers as a relatively unimportant pause in an otherwise busy day.

Notwithstanding, it is largely accepted that the 'free' play behaviours of children can be shaped by the contexts in which they are placed, and the wider geography of the environment, such as the human and physical dynamics of the space [23]. For instance, a previous playground observation study [24] found that increasing the amount of play features on a playground can increase the usage rate of these areas by 5 to 7%, per added feature for boys and girls, respectively. Colabianchi et al. [24] observed 20 recently refurbished urban school playgrounds and predicted that an increase of 10 items on the playground (e.g. sports courts) would increase usage by 50% in boys and 70% girls.

However, previous systematic reviews of school break-time interventions have found both positive and negative effects on physical activity levels [25, 26]. For example, there was no consistent effect in outcomes reported in multicomponent interventions (n = 22) (including teacher training, line markings, staff and student training), or structured break-time interventions (n = 7) (including sports coaches, organised games, PE activities introduced to break-times), with both positive and negative results reported on physical activity levels [26]. Whilst interventions which introduced loose playground equipment (n = 5), though fewer in number, found consistent positive effects on PA levels [26]. However, a recent observation study highlighted that the type of equipment provided can have a negative effect on physical activity levels, such as providing equipment that is too advanced for the children's motor skill ability [27].

Therefore, it is likely the variety of effectiveness reported in the aforementioned studies are the result of contextual, organisational and cultural differences, and the funding and resources available for use during this period of the day.

In the UK, the Department for Education (DfE) provides eligible primary schools with funding from the Primary Physical Education and Sports Premium (PPESP) with the aim of enhancing the health and well-being of pupils [28]. There is a growing amount of support for

the use of the PPESP to enhance children's play and activity by making changes to the outdoor environment [28, 29]. Furthermore, one of the five key indicators aligned to the aim to support the engagement of all pupils in regular physical activity is *'encouraging active play during break-times and lunchtimes'* [29]. Whilst the funding model and its key aims appear laudable and transparent, the extent to which the experiences and viewpoints of children are considered when designing suitable and sustainable outdoor play spaces remains unclear.

When designing a playground environment for children, previous literature indicates that the 'otherness' of childhood ought to be considered [30]. For example, Jones argues that when adults reflect on their own childhood, it is probable that those reflections are 'filtered' by the experiences they have had since their adult becoming [30]. This is not to say these experiences are wholly irrelevant, but they cannot be straightforwardly applied or transferred to children lives today. As previous researchers have suggested, children operate with a different, more flexible and unfiltered negotiation of their world [30, 31]. Previous well-intentioned methods of increasing physical activity in children has perpetuated a "controlling and oppressive way" [32] of coercing children to engage in physical activities. With this in mind, we join a growing list of scholars who argue that many adult prescribed and adult facilitated interventions can be counterproductive [33–35] and call for the meaningful inclusion of children and key supervising staff in the design of childhood play spaces in primary schools.

Notwithstanding, the activities and spaces available to children during school break-time are often designed, chosen, and enforced by adults, leading to the creation of play spaces using the method of "seeing through the child's eyes" [30]. Although there seems to be a genuine attempt from the adult population in the primary school environment to promulgate activities that might be attractive to children [27] there must be some acknowledgement of the 'unbridgeability' [30] between adult and child experiences.

Therefore, the aim of this study is to develop a deeper understanding of user's (children and supervising staff) perceptions of the current playground environment available to children during break and lunch times. Furthermore, to fully comprehend the interwoven and implicit subcultural constrictions and enablers of playground action, child and staff perceptions of the barriers and facilitators during designated physically active break-times are explored. Using the socioecological model, which recognises the interdependent relationship between the intra-personal (individual), inter-personal (social), and the environment, we explore some of the reasons why children engage, like and dislike specific areas of the playground. To the author's knowledge, this is the first socio-ecological investigation of the UK primary school playground environment that has aimed to identify barriers and facilitators of physical activity at each level of the socioecological model. In what follows, we argue that understanding how these factors interact and influence physical activity levels of children during school lunch and break-times can be used by policy makers and individuals in positions of seniority (e.g. head teachers, trusts, funding authorities, local authorities) when planning school playground provisions [16] and in the design of playground interventions.

## 2. Methods

The Consolidated Criteria for Reporting Qualitative Research (COREQ) checklist [36] was used to provide transparency and to ensure accurate reporting of the empirical data [37]. The methods that follow are a brief overview of the procedures used in this study.

### 2.1 Recruitment

Following ethical approval to conduct structured interviews with adults, and focus group activities with children in primary school educational settings (School of Health and Life Sciences

**Table 1. School demographics.**

| | Children on record (n) | Female/male (%) | No. of focus groups per school | % free school meals |
|---|---|---|---|---|
| School A | 565 | 51/49 | 2 | 47 |
| School B | 520 | 49/51 | 3 | 49 |
| School C | 303 | 52/48 | 4 | 68 |

ethics sub-committee at Teesside University, Application Number: **250/18**), five schools from the lowest 10% on the Index of Multiple Deprivation (English indices of deprivation: Department for Communities and Local Government) from the Tees Valley in the North East of England were contacted via email. Schools were selected using the list of local schools (www. gov.uk) and were initially chosen for convenience of location and their urban setting. Schools were eligible to take part if they had a minimum of one year five and one year six class. Schools which matched this criteria were then contacted with details of the study. Four schools returned expressions of interest and were contacted further to discuss the project requirements and complete the school management consent forms. Head teachers from three schools (Table 1) returned managerial consent. Study information and the relevant consent forms were provided for eligible staff, parents of eligible pupils and pupils themselves (assent forms). Staff consent and pupil assent were completed immediately prior to data collection.

## 2.2 Participants

School staff that were in an active role within the playground or in physical activity promotion within the school (PE specialist, health leads, heads and assistant heads, school classroom teachers, playground supervisors and school sports coaches) and children from years five and six (9 to 11 years old) were eligible to take part.

Children took part on focus group activities which were conducted over the course of one school term. Staff were given the option of participating in a face to face structured interview or completing the interview asynchronously using an open-ended questionnaire format. Table 2 reports the number of staff and children recruited in the study.

## 2.3 Data collection

Data collection activities were conducted by members of the research team (MG and AI) that had previous training and experience of working with primary school aged children in a both a prescriptive (teaching and coaching) and facilitative role (research activity). The focus group topic guide is included in the S1 File.

**2.3.1 Data collection–child.** Focus groups, inclusive of a number of data collection activities has been used as an effective way to gather opinions and experiences of the children [38]. Focus groups are a rich method of revealing attitudes, experiences, and perceptions of the target audience [39]. They are particularly useful in the early childhood context, providing an effective means to involve children in research that they are directly implicated in and by. The

**Table 2. Number of staff and children recruited.**

| | Male | Female | Total (n) |
|---|---|---|---|
| Staff (n) | N/A | N/A | 11 |
| Pupils (n) | 31 | 34 | 65 |

Abbreviations: N/A = not available.

research team used a variety of age-appropriate data collection techniques, such as visual prompts, mapping techniques and drawing activities. Techniques such as these, can spark children's interest and maintain concentration whilst providing opportunities for children to more effectively engage with the task [40–42].

The focus group discussions were designed to last for a maximum of 60 minutes, as recommended by previous research on the topic [38] and were conducted in a segregated, quiet, informal space within their familiar school environment. To be confident in successful data collection and provide a positive experience for the children we limited group size to eight children [41]. At the start of each focus group, children were welcomed and introduced to the focus group facilitators and read the summary of the information previously provided to them. School staff were not present during the child activities as it was felt that the presence of these authoritative figures might have affected the honesty in responses.

All focus group discussions were digitally recorded using audio devices. Noticeable changes in body language or persistently repeated opinions were recorded in the facilitator notes to aid in transcription, to support the outputs from the variety of focus group activities and to ensure accuracy of the adult perception (i.e., the research team) of the child's experience/response. Children were told the devices *"are here to record the discussions we have today about your playground. They will only be used by the research team and the recordings will not be shared with anyone else"*. Nevertheless, there were a few occasions where the children sought confirmation of anonymity; *"will my teacher see this*?" This highlighted the power imbalance between adults and children within this setting. Reinforcing the anonymous nature of their responses and explaining that the role of the facilitators in this activity was to listen to their experiences and stories and not to judge or discipline the children [38] served the purpose of removing any anxiety of 'getting into trouble' for speaking openly, and reduced the inherent power imbalance between adults and children with the facilitator from this point on being perceived by the children as part of the group, maximising interaction and honesty.

During transcription, individual responses were coded by participant number only (e.g., pupil 1, pupil 2 etc.) through the recognition of a change of voice. Focus group activities continued until facilitators believed the groups had reached a saturation point [43], at which point the subsequent activities described below were introduced.

The first task required children to highlight on an A3 map of their playground, areas they liked and disliked, and provide reasons for their responses (Fig 1). Mapping techniques and visual prompts have been identified as an innovative and useful way for children to express their views about the use of the spaces they occupy [38, 42]. Children were encouraged to be creative and draw on the maps if they wished. This activity was designed to gain a wider contextual understanding of the children's perception of the playground environment [16].

Children were then asked to write the skills they perceived as necessary to use each playground area on sticky paper notes and place them on the map over the corresponding playground area. On completion of this task, children then removed the sticky notes and placed them in a line from the most to least important, in terms of being able to use the playground effectively (Fig 2). The outputs from this task were used to identify any specific skills that children perceived as necessary to be able to be physically active in each of their previously identified playground zones.

For the next task children were given an A4 piece of paper and a selection of pens and pencils and were asked to draw the image that came into their head when thinking of a playground supervisor (Fig 3). Previous studies exploring the effect of the management and supervision of playground activities on the level of PA during break-times have found contrasting results [27, 44, 45]. This suggests that the roles, actions and behaviours of playground supervisors can have either positive or negative connotations on children's behaviour [46]. Using creative

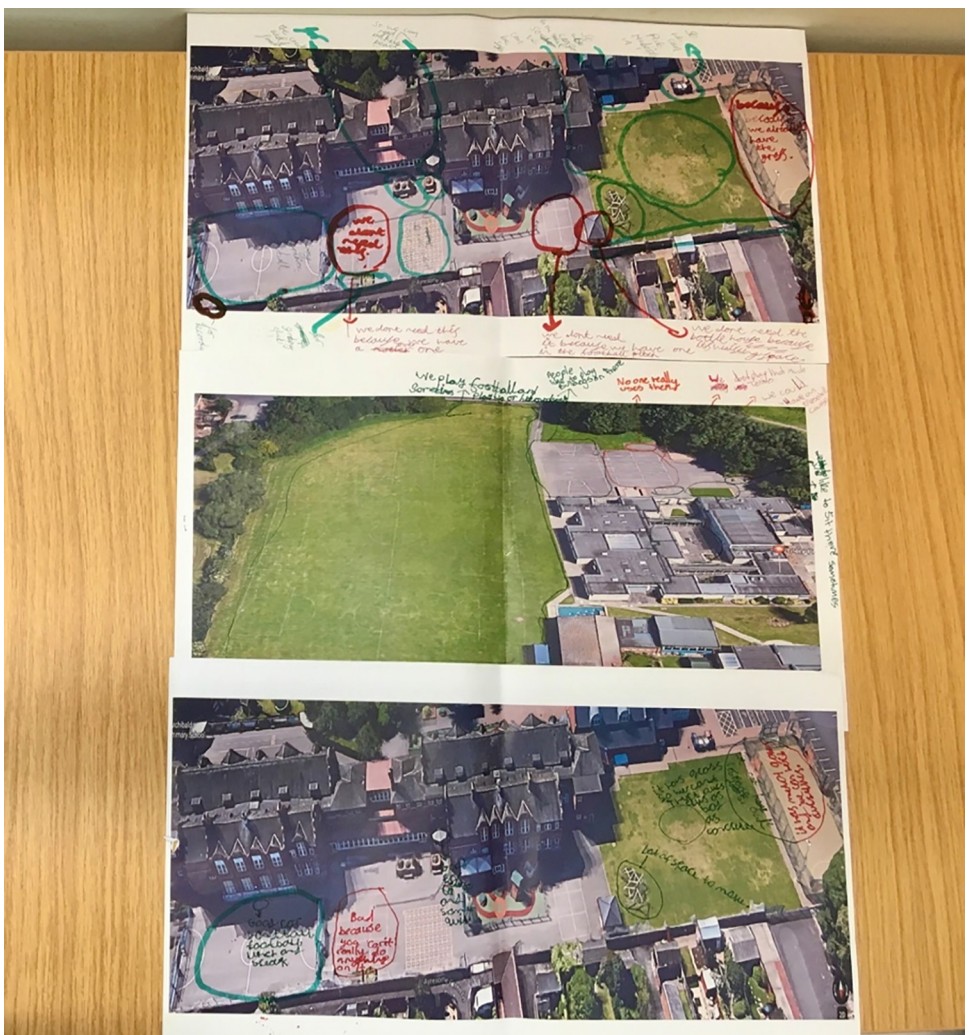

**Fig 1. Aerial playground mapping activity.**

approaches, such as drawing, can be more effective in interpreting a child's perception of their experiences and emotions [47]. To help them get started with the task children were first asked to write some words down that described their experiences of playground supervisors in their school playgrounds.

The final focus group activity was designed to allow children complete anonymity and remove themselves entirely from the confinement of restrictive adultist opinion. Children were given one piece of A5 paper and asked to *"write one wish for the playground that would make it better and help you be more active during break-time"*. The children were then asked to fold their piece of paper in to a small square and post it into *'the secret box'*. Previous work has suggested a 'secret box' activity removes the fear children have of sharing their thoughts and opinions [40].

The facilitator's role in these tasks was to look for clarity in the responses and activity outputs, to ensure the children had considered each of the task requirements, and stimulate further discussion amongst the group. The discussions were used to get more accurate interpretations of the outputs during audio transcription.

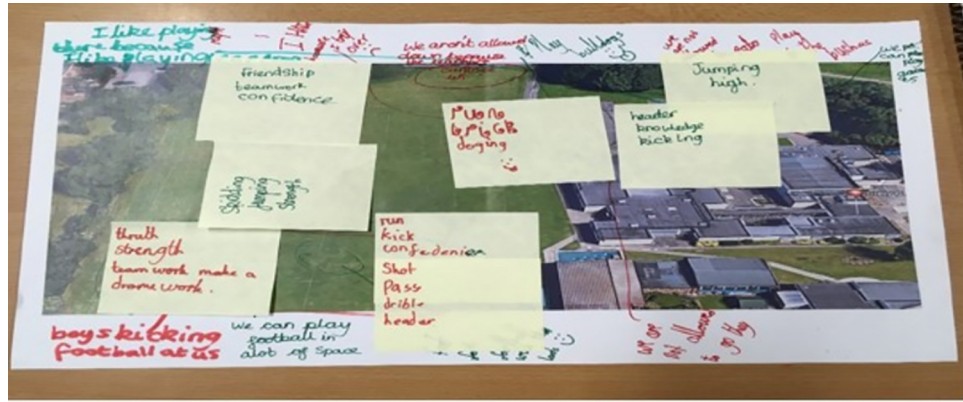

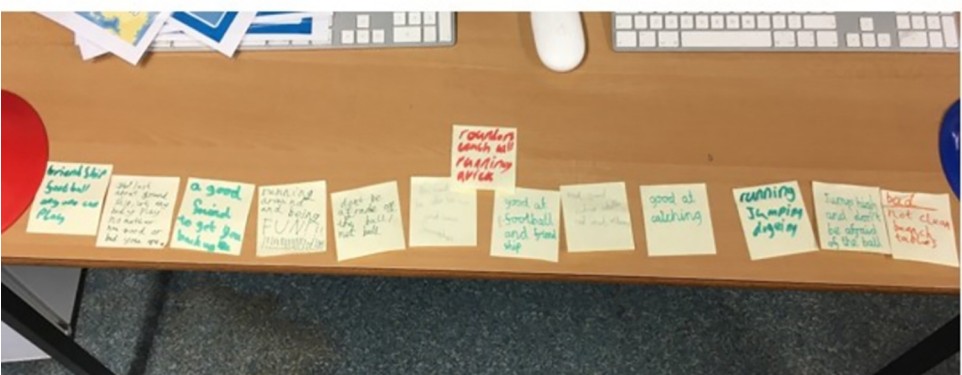

**Fig 2. Skill requirements of playground areas and order of importance.**

**2.3.2 Data collection–staff.** Staff were first offered a one-to-one interview to discuss the a priori themes of the project; *barriers* and *facilitators* to a physically active playground during school break-time. However, gatekeepers at each of the schools expressed a concern from teachers on allocating time from their day to meet with the researcher. Furthermore, there was a concern that senior leaders at the school would be able to identify who had and hadn't taken part in the project. For this reason, staff were given the option of interview or questionnaire. All participating staff chose to complete the questionnaire in their own time and were asked to be as detailed as possible in their responses on the questionnaire, using additional pages if needed. Staff were offered the option of providing contact details if they were happy to be contacted further for any responses requiring clarification. No member of staff provided these details.

## 2.4 Thematic data handling and analysis

Qualitative data for both focus groups and asynchronous interviews were analysed using the process of thematic data analysis as described by Braun and Clarke [48]. This 6 stage approach allows a more detailed contextual examination of the pre-identified ideas, assumptions, and ideologies underlying these a priori themes without sacrificing its flexibility to provide "a rich and detailed, yet complex account of the data" [48; pg.5] that is both theoretically and methodologically sound; and can be widely used across a range of epistemologies and research questions [49].

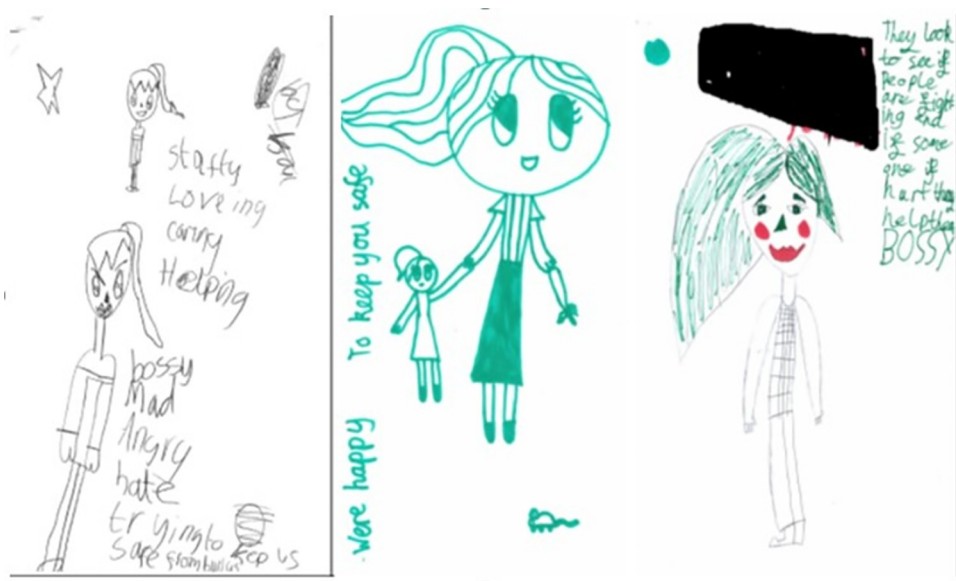

**Fig 3. Playground supervisor drawings.**

**2.4.1 Child focus groups.** The first task for data analysis involved two researchers reading through every focus group activity the children had completed to begin to identify recurring themes across each of the groups (stage 1—familiarisation with the data). Each activity was then reviewed again, and initial features of the data coded in a systematic fashion to collate data relevant to each code (stage 2 –generation of initial codes). Activities were reviewed a third time whilst listening to the associated audio recording from the matched focus group to ensure the children's written points had been interpreted accurately.

Audio recordings were not transcribed "verbatim" but were used to ensure that valuable detail relating to the context and the specific nature of the written responses were captured [50]. Exerts from the audio recordings which matched and supported the focus group activity outputs were transcribed verbatim (by each researcher) and transferred to the table of responses and coded accordingly. As codes were collated, potential themes began to emerge and all relevant codes (and associated data) were transferred under these themes (stage 3 – search for themes). On completion, themes and the associated data items (audio transcriptions and written text) were then reviewed to check for accuracy of interpretation and for any repetition across themes (stage 4 –review of themes).

The latent themes that emerged as a result of the aforementioned analysis were grouped under the component titles of the socio-ecological model; individual, interpersonal, physical environment and policy (stage 5 –Definition and names of themes). The multi-level framework that the socio-ecological model provides, allows for a constructionist and interpretative examination of the range of socio-cultural factors that can influence physical activity levels during school break-times [16, 48]. This final activity facilitated the creation of the final thematic map and the final interpretive report (stage 6 –production of report) [48].

**2.4.2 Staff asynchronous interview forms.** Completed staff forms were read in full prior to analysis to identify commonality across all responses and to become familiar with the data. Data was then coded and handled following the same processes described above. Responses from the child focus groups and staff responses that did not recur frequently but that had

particular resonance due to the language used were grouped under the same code ('valuable insight').

## 3. Outcomes

A total of 65 children were recruited and provided parental consent and initial assent. At the time of data collection four children were absent and three withdrew assent prior to the start of the focus groups. The remaining 58 children (52% female) participated in focus group activities.

Eleven members of staff from across the three schools returned consent to take part in the study. Figs 4 and 5 display the final thematic map for children and staff, respectively. The thematic map is inclusive of the a priori themes (barriers and facilitators) and the deductive themes from each of the data collection activities (playground map, essential skills, supervisor drawings and discussions) and their association to each of the socio-ecological model components. The secret box activity was also analysed in respect to the socio-ecological model but did not contribute the themes identified in the thematic map.

A full list of the secret box responses can be seen in Table 3. Responses are separated for children and staff and divided into small and large wishes dependent on the resources (physical and monetary) needed or the surface area required [16]. Further, the wishes are separated into categories based on their desired outcomes (i.e., physical environment, individual/interpersonal or policy). Children's wishes focussed on play, adventure, and fun. Wishes were predominantly concentrated on the provision of new equipment and longer break-times. Staff wishes for the school playground focussed on a wider development of playground structure, policy changes, management and support.

## 4. Discussion of research findings

The following is a further presentation of focus group outcomes, discussed in light of theory and research on the individual, interpersonal, environmental and policy influences (barriers and facilitators) on children's physical activity engagement during school break and lunchtimes and is subdivided according to the components of the socio-ecological model. The style

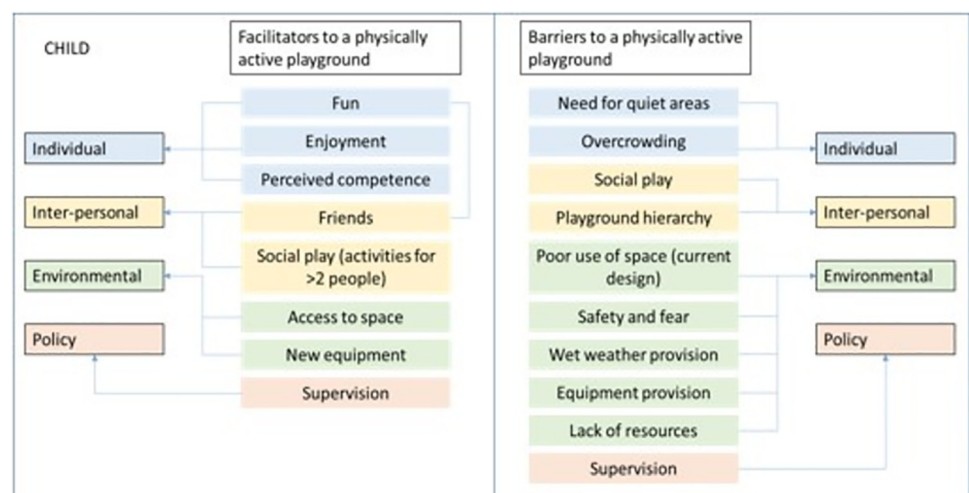

**Fig 4. Final thematic map showing the socio-ecological barriers and facilitators to a physically active playground from the school children's perspective.**

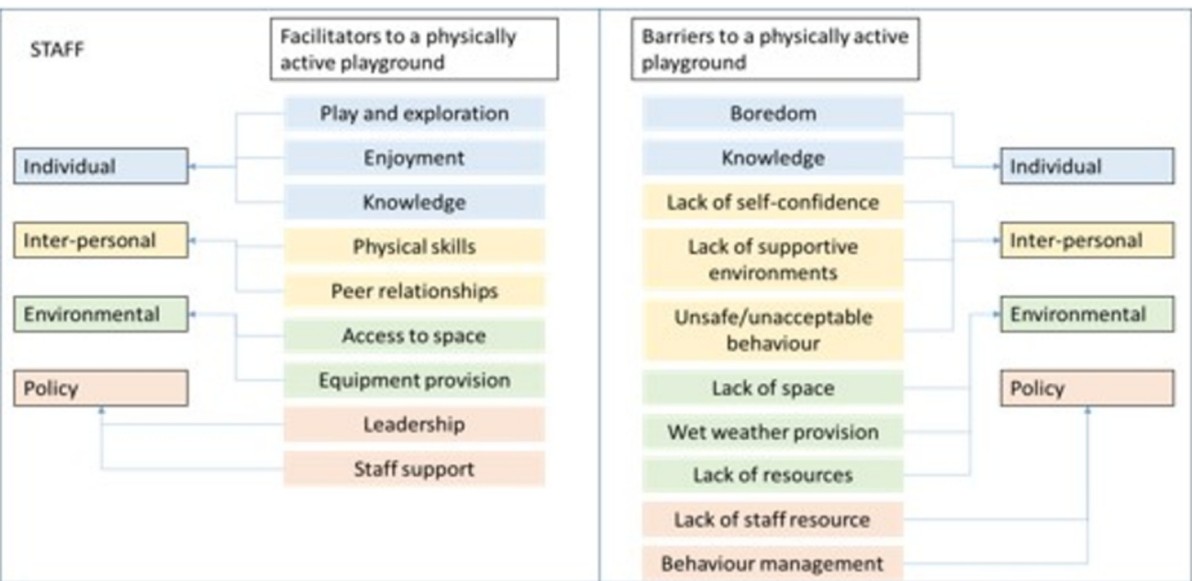

**Fig 5. Final thematic map showing the socio-ecological barriers and facilitators to a physically active playground from the school staff perspective.**

of the discussion that follows is in contrast to previous qualitative investigations as the authors present a critical discussion interwoven with pupil and staff transcriptions from the data.

## 4.1 Individual and interpersonal factors (children)

From the perspective of the children, individual level facilitators of physical activity focussed predominantly on the intrinsic desires to have fun (*"Because my friends push me on the low swings, it's fun"; "it is fun to try new things"; "me and my friends play games here. . .the maze game because it is fun"; "we play tag, it's very fun"*), for the enjoyment of activities (*"I like it because I get to play football"; "I like playing there because I can play leapfrog"; "I like it cause we can play tennis and get tennis rackets"*) and the belief they will do well in a specific activity (perceived competence) (*". . .football is a good sport for me"; ". . .because I likely do well"*).

Thus, congruent with the work of Snow et al. [35] who conducted focus groups on 8 to 10 year old girls, children in this study cited fun, physical competence and mastery of skills as a major influence on the engagement in play. In terms of physical competence, Barbour [51] suggested that the type of activities children take part in are a result of similarities in movement ability and movement skill competency, with children of low physical competence reluctant to approach activities requiring a higher level of ability. Evidence suggests that when fundamental movement skills are taught to younger children (4 to 9 year olds), increases in confidence in their ability results in participation in physical activity during other parts of the day [52]. As children age they are more aware of their ability, or lack thereof, and as a result less likely to participate in activities they desire for fear of embarrassment [52, 53]. The desire for actual physical competence in Snow et al. [35] and the engagement (or disengagement) in specific activities due to perceptions of physical competence in this study are slightly difference concepts. However, the aspirations for and perceptions of competency were driven by the same yearning for a sense of social belonging.

Children in this study identified that they took part in activities that they *"would likely do well at"* but also participated in activities and occupied playground areas for social reasons, irrespective of any assessment of physical competence and in the absence of a specified activity

**Table 3. School children and staff magic wish responses.**

| | | Children | Staff |
|---|---|---|---|
| Physical environment | Small items | • Cargo nets | • Scooters and bikes |
| | | • Monkey bars* and gym equipment | • Be able to use the grass |
| | | • Slides* | • More equipment* |
| | | • Swings* | • New fresh games |
| | | • Seesaw | |
| | | • Tyres | |
| | | • Bikes and scooters* | |
| | | • More equipment* | |
| | | • Make it more fun | |
| | | • Trampolines* | |
| | | • Something fun–like hunts | |
| | | • Fairer games | |
| | | • Spider net climbing frame | |
| | Large items | • Obstacle course | • A school field for summer to avoid confrontation |
| | | • VR booth | |
| | | • More options for indoor play* | |
| | | • Climbing wall with buzzers | • New grassy area |
| | | • Running track | |
| | | • A field so we can do rugby | • Overhaul of the outside area—more engaging |
| | | • Make a basketball pitch | |
| | | • More playground things | • MUGA on the concrete area (less injuries) |
| | | • Swimming pool | • More outdoor to explore |
| | | • Big massive slide | • A more interesting environment to explore |
| | | • Big Bouncy castle | • More space |
| | | | • A sheltered area |
| Individual and Social | | • I wish to make everyone happy on the playground | • Self-regulation |
| | | • More exciting games with more people | • Personal power and resilience–to cope with losing |
| | | • Do dangerous stuff | |
| Policy | | • More options for indoor play* | • Training for staff* |
| | | • More time* | |
| | | • Tag rugby coach | • Training for playground leaders* |
| | | • Less tolerance to bullies | • More equipment |
| | | • More time on the ball-court | • Involve staff more |
| | | • To be able to use key stage1 (5 to 7 years old) playground* | |

*Items occurred multiple times in magic wish responses (multiple = three or more).

("*this is where my friends are*"; "*because most of my friends play here*"; "*because my friends are here. . .*"). Parrish et al. [52] focus group findings from children aged 9 to 11 highlighted that children were more likely to take part in games their friends were playing, even if they had a desire to play something else. One group of children in this study, when discussing the activities on their playground highlighted:

> "*let everyone take part and be nice, we don't really care what skills you have we just like letting people play, it's just about friendship*"

Previous research suggested that there is more than a simple gender preference operational when children select areas of the playground to "play" in [27]. The influence of physical competence, perceived physical competence and friendship identified here, re-enforces this assumption and highlights the potential impact of positive peer relationships and social position as a driver for physical activity engagement. During analysis, there was a clear interaction between findings at each level of the socio-ecological model. However, this interaction was particularly evident between the individual and interpersonal items in the model. Many of the 'individual' factors children gave for liking and disliking areas were driven by the desire for 'social' interaction or 'social' play. For example, the individual desire for quiet and relaxation (*"I like it because it's a good place to private talk"*) and for playing games (*"me and my friends play games here"* and *"I play tag with my friends"*), were grounded by positive peer relationships.

The desire for children to engage in social games, requiring more than two people could be perceived as a method employed by the children in this study at increasing the 'quality' of their friendships

> *"because we get to run around and play bulldogs"*, *"we sometime get to play football tennis"*, *"we play football and sometimes tig"*, *"we play hide and seek"*.

However, the opportunity for social play was also often linked to less desirable playground experiences (*"there are too many footballs"*, *"there are a lot of fights and it stops playing"*, *"play is too rough"*) and traditional playground hierarchies (*"the boys take the ball court most of the time"*, *"because other year groups use it"*, *"a lot of fights with year 6's")* which could be considered as barriers to physical activity for individuals who avoid competitive games for fear of conflict and to avoid the hegemonic masculinity [54] of the sporting (predominantly football) culture of the primary school playground [55].

Hegemonic Masculinity is a term popularised by sociologist R.W. Connell [54], to explain the recurring socially constructed practices that promote the dominant social position of males and the subordinate position of females across the life cycle. Within a school playground environment children can become invested in activities that help them to construct and maintain a gender identity [56, 57], with the environment and their peers often enabling their construction of 'masculinities' and 'femininities' [57]. Although the presence and the magnitude of the effect of these practices are likely school dependent, due to the varied management/supervision of the playground between schools, football and fighting is an activity that many boys continue to use to solidify their masculinity [56].

Some of the girls in this study identified hegemonic masculinities that are displayed during break-times. As an example, the following conversation between three participants is worth citing at length. On this occasion we use pseudonyms to enable the reader to distinguish between participants.

**Pupil 1**: *"we don't like playing here because you get hurt and the boys kick the footballs at you" (Girl)*

**Pupil 2**: *"there is loads of fights" (Girl)*

**Pupil3**: *"No..." (Boy)*

**Pupil1**: *"YEAH THERE IS" (Girl)*

**Pupil 2**: *"have you seen how many fights happen" (Girl)*

**Pupil 1**: *"there was a fight here" (Girl)*

**Pupil 3**: *"oh yeah there was a fight there the other day" (Boy)*

**Pupil 3**: *"we have fights constantly" (smiling) (Boy)*

**Pupil 1**: *"I hate it" (Girl)*

One boy in this group can initially be observed trying to address these statements by perhaps claiming either the absence of fights or trying to explain the reason for fights, before he is interrupted. He then concedes and becomes somewhat proud with a contented claim of *"WE have fights constantly"*. Whether he actively participates in this behaviour or not, this statement could be perceived as attempt to associate himself to these hegemonic masculine behaviours deemed important for his social status.

Football has been [55] and continues to be [33, 58] the predominant activity dominating playground space. Similarly, the schools participating in this study had playgrounds which were monopolised by the established football space (marked and worn out pitches, caged football zones, painted goals on walls). The domination of the playground space for football leads to a desperate rush by children at break-times, to preside over the remaining playground space. Thomson [33] observed children claiming possession of playground space by marking areas with their coats and school bags for their activities and any attempt at invasion from others resulted in retaliation and conflict. This issue becomes exacerbated during winter months when access to the play spaces hosting these dominant playground games is prohibited, directing these activities into the already contested areas of the playground.

## 4.2 Individual and interpersonal factors (adult supervisors)

Similar individual level facilitators were identified from staff outputs with play, exploration and enjoyment identified as key to children's participation in activities.

*"Children like to climb on the rocks and tyres"; "children often look to play their own games. . ."; "children like freedom and unstructured play"; "children enjoy playing football"; "children enjoy the ball court and playing football"*

Although adults (staff) in this study seem to understand the individual value of play, they identified more frequently with the extrinsic values of peer relationships and social development:

*"teamwork and collaboration"; "ability to listen to others"; ". . .take turns and play fair", "need to understand the rules"; "social is important to feel comfortable playing in front of others"*

Previous research exploring children's geographies has highlighted that the intrinsic value of play is not acknowledged by teachers and policy makers [35] and that opportunities for play, particularly outdoor play is decreasing with increased emphasis on classroom based, adult organised activities [59, 60]. Furthermore, adults colonise children's places and create safe and easy to monitor play spaces which often means the naturally sporadic and exploratory play behaviours of children [33, 61] are perceived as disruptive and undesirable, and are consequently dealt with 'accordingly' (*"children need to be guided on how to play safely"*, *"children need to be aware they will be punished (equipment removed) for bad behaviour"*).

The staff opinions on the 'correct' use of the playground could be interpreted from a dualist perspective, whereby there is either a right or wrong way of 'playing'. Although one cannot argue that children will benefit from *"teamwork and collaboration"* and an *"ability to listen to*

*others"* throughout their child, adolescent and indeed adult becoming; the adult regulation and enforcement of these qualities goes against the nurturing concept of physical literacy [62]. Children develop a natural, more flexible interaction with the environments that surround them and can be very creative and innovative when adapting architectural features of the playground such as bins, bollards, fencing, walls etc. [33]. Objects in the environment are not inanimate features to which we ascribe an abstract concept but are meaningful in a sense that they 'engage' with us, indicating how we can interact effectively with them [63]. Children in this study identified areas of the playground that to the researcher looked unusable. However, children circled these areas for the inanimate objects (bollards, rocks) that existed there (for example, *"I like playing here cause I can play leapfrog"*). However, these behaviours are often stifled by staff on the playground perceiving their use as inappropriate and unsafe (*"children given free choice often decide on inappropriate games"; "children need to follow the rules and understand what they can and can't do"*), and because they do not fit in with their framework of rules. Jones [30] suggested adult constrictions, desires and agenda restrict children's lives and their practices when discovering their identity in a changing environment.

Children learn very early on the notion of rule keeping and are generally faced with a daily list of 'don'ts' before entering their play space [33]. Crease [64] explains that infants go through a number of stages in their becoming, described as first 'I move', then 'I can', and finally 'I can do'. A large proportion of children in this study were faced with physical barriers, boundaries and rules which reduced their freedom to 'move' and therefore unable to explore the 'I can' and subsequently the 'I can do. . .'

## 4.3 Environment and policy level

As previously mentioned, the large open spaces identified in this study were predominantly grass fields and expansive concrete areas. Children highlighted these areas as positive for their promotion of team games, playing with friends and their soft surfaces. However, the children also highlighted that these areas often flood in wet weather leading to prohibited access due to adverse conditions. The data from the children and staff suggest that this is an issue that needs addressing at policy level with adequate investment in facilities for all weathers:

Children

*"sometimes not allowed here when it is wet or muddy"*, *"can't use it when it is full of snow"*, *"not allowed in when it is snowing"*, *"we are not allowed on the grass when it is wet"*, *"we are not allowed on when it is icy or snowy cause we might fall over and get hurt"*, *"when it rains there are puddles for weeks"*

Staff

*"space is a problem when the grass is wet, children are confined to the hard area which prevents children playing"; "bad weather prevents physical activity at break times"; "not being able to use the field when it is wet has a negative impact as children are not allowed footballs on these days"*, *"rock area is dangerous when it is wet"*

As one child said "*if it is raining, why not put a roof on the MUGA"*. Similar findings from Australian children, also recognised the need for 'weather protection' [65], demonstrating that despite very different weather conditions, the play restrictions being enforced on children in primary school playgrounds is an issue experienced internationally.

The appearance of staff members on the playground acting like shepherds tending their disobedient flock may be driven more by the inadequate investment at a policy level in the

children's physical, social and emotional development during this important period in a child's day [59]. This was further highlighted by a number of staff members who identified a lack of staff resources prevented them from engaging in anything other than crowd control (*"there is lots of activity and a lot to monitor for just two members of staff"*, *"not enough staff being able to supervise and keep children safe"*, *"staff are limited, we already have some staff on the playground but not all the time and they can't cover everywhere"; staff are occupied dealing with behaviour so seldom able to engage with activities"*). This is in contrast to self-report findings from national (UK) school surveys from 1995 to 2017 which identified that there are now more adults supervising than there has been in the previous twenty-two years [59]. Although these numbers are likely school dependent, the actions of the supervisors may be more important than the numbers available [27]. Children highlighted the potential for teachers to act as facilitators (*"some teachers won't come out but 'Miss D' played like Mr Fox or something with us before but not many (teacher) do")* but are too often restricted by the number of staff available (*"sometimes there is only one member of staff on duty so we have to stay where the teacher can see them so they are safe and don't get hurt"; ". . ..but I do get it cause there are only like two dinner nannies"; "that's the part we are not allowed down, well we are sometimes but not all days when we don't have teachers, because when it (the bank) goes down the teachers can't see us"*).

Children at participating schools had a mix of teachers, teaching assistants, 'dinner nannies' and 'playground friends' that helped monitor the playground during break-times. Children highlighted they would like their teachers to be more involved during break-time but highlighted they wanted teachers based on 'sportiness' (*". . .because they are good at sport"*, *"teachers are not that sporty"*, *"Mr T and Mr L are the sportiest but there are no more sporty ones"*, *"our teachers are not that sporty, there is only like three and they are not that sporty"*).

Although in the current study we were unable to distinguish between staff positions within the school (head teacher, teacher, teaching assistant etc.) due to the anonymous nature of the staff responses; previous research has found that head teachers from different schools have very different ideas about the value and role of break-time [20] and therefore, the behaviours, actions and opinions of the staff (from staff and child perspectives) in the current study may be a result of (or lack of) the agenda at senior management levels.

Overall, staff perceived their role as a combination of encouraging a supportive and safe environment (*"supervisors should be at their station, organising resources and facilitating"*, *"adult presence ensures that children feel safe and are used for advice and support if needed"*) and promoting engagement in physical activity (*"my role is to keep children safe and happy and to encourage some children to be active"*). However, the perception from children was that the role of playground staff is for safety and the enforcement of rules and boundaries (*"sometimes we do use here for bulldogs, but the younger ones are doing it now so we are not allowed"*, *"if we go on there the teachers can't see us and we'll get dirty"*, *"dinner nannies say we can only play with your own year group. . .it's so annoying. . ."*, *"they look after us, stop fighting and help people who are hurt"*).

The active interest of the adult members of staff in the school were explored during the playground supervisor and playground activities tasks. When asked about the staff who were present on their playgrounds during break and lunch-times, this is just a sample of the words the children used to describe them:

*Safe; loving; try to keep us safe from bully's; caring; angry; helping; laughable; sharing; bossy; hardworking; respectful; kind; mad; safety; hate.*

Although mostly positive, the variety of qualities cited by the children gives an idea of the variety of adult personas that occupy children's playgrounds during break-times. It is therefore

important that these staff members understand the importance of their behaviours and the positive influence they can have on the social and physical activity behaviours of the children who occupy the playground space.

From the variety of staff and child accounts provided in this study and in previous studies [16, 33] on the level and role of staff interaction during break-times it seems that, beyond child safety, there is no standardised, universally accepted requirement for behaviour of playground staff in the primary school setting. This allows for a large variation in the day to day management of the school playground, dependent largely upon the member of staff who happens to be 'on duty' that day (mood, personality, personal agenda, etc.).

All the schools in this study were in receipt of the PPESP. Only one school in this study mentioned break-times as part of their planning, with structured lunchtimes with a sports coach, lunch supervisors and pupil leaders to target inactive children during break-time reported. Whilst this one school's acknowledgement of break-times as a period of time that would benefit from investment, the plans and ideas mentioned previously were alongside the provision for the daily mile, access to new sports and activities and a lunchtime wake up dance activity–all of which was allocated a combined £750 from the £19,520 PPESP allocated to this school.

Whilst the physical activity levels of children during break-times is much more complex [66] the lack of valuable and sustainable investment in playground provision is worrying and in contrast to the recommendations provided by the DfE [29]. A continuation in the marginalisation of break-times for more curricular focussed adult led activities (i.e., PE); alongside a reduction in time provided for break-times [20] and inadequate investment in the primary school playground provision, will lead to further reductions in exploratory play and reduced opportunity to develop physical literacy. Furthermore, without recognition of the importance of break-times in children's physical, social and emotional development and the provision of a sustainable intervention, the current playground behaviours will continue to re-enforce the adult-child power distribution [30].

## 5. Implications for practice

This study aimed to use the socio-ecological model to explore school children's and school staff perception of the school playground and identify reasons for enjoyment, engagement and dissociation with specific playground areas. There have been limited studies exploring the socio-ecological model components within a school context [16] and to our knowledge this is the first use of this framework to qualitatively explore the complex contexts presented to UK primary school children during their 'free play' time.

This qualitative evaluation has identified differences between the adult and child perception of the primary school playground. These differences affirm the need to actively include children in future playground planning. Many schools ask their pupils *'what should we do?'* Or *'what would you like on the playground?'* However, for most, this is where this partnership ends. This does not go unnoticed by the children who have invested a part of themselves in these tasks *("the teacher said we could get like a science area outside to grow plants and things but she never did it. . .I don't know why")*. It is important to follow up on these activities and feedback to the children on the actions been taken, even if the outcome may be perceived as undesirable, so that they feel that their opinions are heard and of value [41].

This somewhat unconscious stance of power and knowledge is often overlooked in environments where the focus is on making well-intentioned changes to the environment 'for the children's sake'. However, the issue still remains and we, as adults know little about the child's becoming and cannot accurately see things from a child's perspective [30].

Effective injury prevention efforts at school are important and should address several factors (i.e., Individual, interpersonal, environmental and policy). However, improvements to the physical environment of the school through regular safety assessments, good quality maintenance, and repairing hazards immediately after they are identified [67], can contribute to the safety of the school children without the need to restrict children's access to specific areas. Although the safety of children should be paramount, children should also be allowed some freedom to choose the activities they wish to take part in, to be able to begin to explore the concept of becoming physically literate. Physical literacy, focuses on the lived body, the embodied dimension of human existence [63], therefore nurturing this aspect of children's lives will make a distinctive contribution to their becoming.

As mentioned previously, football dominated the playground, monopolising the space available. Cashmore and Dixon [68] explain that football is inescapable, a sport ingrained into the fabric of communities. It would seem that this is also the case within primary school playgrounds, where football remains the activity dominating the available space. Therefore, as many children engage in this activity during break-times, it can be considered an important and effective catalyst for physical activity participation. However, the barriers that this dominance presents to children, either not interested in football or who have yet to demonstrate an acceptable skill level, cannot be overlooked [33, 58, 69]. Conversely, as previous focus group studies with children have suggested, it is the lack of alternative space that is the main concern [69] and removing the facilities for football would remove opportunities for the large numbers of children who currently use football as a means of being physically active. Therefore, provision of additional space and/or more effective use of the current space, alongside more inclusive and enjoyable activities for boys and girls is needed.

## 6. Strengths and weaknesses

The findings from this qualitative evaluation provides an opportunity for primary schools which match the description of the schools participating in this study, to reflect on primary school playground strategies and practices that are implemented at policy level. However, this study is not without its limitations. Firstly, restricting recruitment to year five and year six children may have overlooked the barriers that exist in the younger key stage 2 children, particularly 7 year olds, who will have just been introduced to this new playground. This limits the ability to generalise this study's findings to children of different age groups who are likely to have a different playground experience. As this study did not receive any funding there was a limit to the number of schools and participants the research staff could manage in the time frame. However, limiting the sample to two year groups from four schools allowed for a more comprehensive data acquisition, evaluation and synthesis.

Regarding the concept of the 'adult filter', the authors of this study cannot remove their own subconscious adult filter and adult embodiment; however, the comprehensive, flexible and robust methods employed during child focus groups, in addition to the use of respondent validation techniques is a strength of this study and minimises any inaccuracies in the adult interpretation. Furthermore, the use of two authors throughout data collection, transcription and analysis enhances the trustworthiness of the findings presented in this study. In addition, due to staff concerns with interviews (mentioned previously) all staff responses were completed using questionnaires, limiting a more in-depth investigation of the answers provided. It is hypothesised that a more comprehensive response and discussion would be possible using interview methods, and every effort should be made to remove the barriers perceived by members of staff in this study in future studies. Finally, this data was collected prior to the COVID-

19 pandemic. During the pandemic in the UK, primary schools changed the structure of break-times, increasing the number of breaks for fresh air throughout the day. This change in structure might have affected children and staff perceptions of the value of break-times. Future research should explore the effect the COVID-19 pandemic had on the perceptions of school break-times during and post pandemic.

## 7. Conclusion

Playground physical activity during break-times appears to be affected by a number of variables at each level of the socio-ecological model. For example, playgrounds which promote a wide variety of activities are likely to encourage higher levels of MVPA. However, individual (likes/dislikes), social (number and quality of friendships), environmental (available space) and policy level (staffing/resources) constraints determine children's activity choices and physical activity levels during break and lunch-times. Exploring and understanding the socio-ecological determinants of playground physical activity during school break-times is an essential part of designing, conducting and evaluating complex environmental interventions. By attempting to understand the effect of the various complex interactions that exist within primary school playgrounds will help raise awareness within schools of the implications of supervisory interactions, judgement and management of behaviour, on the health and wellbeing of pupils [70].

## Supporting information

**S1 File. Focus group and interview topic guide.**
(DOCX)

## Author Contributions

**Conceptualization:** Michael Graham, Kevin Dixon, Liane B. Azevedo, Matthew D. Wright, Alison Innerd.

**Data curation:** Michael Graham, Alison Innerd.

**Formal analysis:** Michael Graham, Alison Innerd.

**Investigation:** Michael Graham.

**Methodology:** Michael Graham, Kevin Dixon, Liane B. Azevedo, Matthew D. Wright, Alison Innerd.

**Supervision:** Kevin Dixon, Liane B. Azevedo, Matthew D. Wright, Alison Innerd.

**Writing – original draft:** Michael Graham.

**Writing – review & editing:** Michael Graham, Kevin Dixon, Liane B. Azevedo, Matthew D. Wright, Alison Innerd.

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
