## [Decision Letter · Decision Letter 0]

2 Nov 2021

PONE-D-21-28206A socio-ecological examination of the primary school playground: primary school pupil and staff perceived barriers and facilitators to a physically active playground during break and lunch-times.PLOS ONE

Dear Dr. Graham,

Thank you for submitting your manuscript to PLOS ONE. After careful consideration, we feel that it has merit but does not fully meet PLOS ONE’s publication criteria as it currently stands. Therefore, we invite you to submit a revised version of the manuscript that addresses the points raised during the review process.

We look forward to receiving your revised manuscript.

Kind regards,

Francisco Javier Huertas-Delgado, Ph.D.

Academic Editor

PLOS ONE

Journal Requirements:

2. Please include additional information regarding the survey, questionnaire, or interview guides used in the study and ensure that you have provided sufficient details that others could replicate the analyses. For instance, if you developed a questionnaire as part of this study and it is not under a copyright more restrictive than CC-BY, please include a copy, in both the original language and English, as Supporting Information.

Reviewers' comments:

Reviewer's Responses to Questions

**Comments to the Author**

1. Is the manuscript technically sound, and do the data support the conclusions?

Reviewer #1: Yes

Reviewer #2: Yes

2. Has the statistical analysis been performed appropriately and rigorously? 

Reviewer #1: N/A

Reviewer #2: Yes

3. Have the authors made all data underlying the findings in their manuscript fully available?

Reviewer #1: Yes

Reviewer #2: Yes

4. Is the manuscript presented in an intelligible fashion and written in standard English?

Reviewer #1: Yes

Reviewer #2: Yes

5. Review Comments to the Author

Reviewer #1: General comments

First congratulate the authors for the research, in my view the theme is extremely important and it takes empathy to transcend the findings, I judge the relevant research and I am grateful to be able to review this manuscript. I have small comments on points of the manuscript that in my view should be improved. They are:

Abstract

• In this session the authors state "Using the socio-ecological model", however do not characterize such a model, I suggest informing which model the socio-ecological was used.

• The authors mention the total number of children and the chronological age margin, however, how many adults participated in the study? What are your school roles? Teachers? And what is the chronological age margin of adults?

• What were the differences perceived by children and adults?

• What are the environmental limits and factors of school policy that are interrelated?

• The authors inform what the study offers, that is, a perspective of practical applicability. However, the abstract does not point to a conclusion. What the study concludes based on the findings?

• Authors need to insert at least three keywords after the abstract.

Introduction

• Line 104 – Need a reference to end of sentence.

• Line 111 – In the introduction prioritize the passive voice, leave the active voice for the discussion session.

• Final paragraph of the introduction: The authors need to present a hypothesis after mentioning the objective, which they believed before the study began?

Methods

• The authors need to insert in the session called "Participants", the specific functions of each adult who was part of the sample, for example how many were teachers? You need to provide more details in this session such as the age group of adults for example.

Results

• No comments.

Discussion

• The additional results should be in the results session and not in the discussion session. The authors should present all the findings in the results session. Later, in the discussion session should start informing the objective of the study, the initial hypothesis and inform through which results the initial hypothesis was proven (or not). Thus, a discussion should be held with the existing data in the literature.

Summary and conclusion

• Set the title of this session only to “Conclusions”.

• The conclusion should answer the objective of the work. All the rest (limitations, observations, views, suggestions, future prospects) should come to the end of the discussion session.

• This session is very extensive, sift the conclusion of the study to a maximum of ten lines in a single paragraph.

Reviewer #2: The authors present a document that is ethically and academically acceptable. They use the robust theoretical framework with some expertise. Even so, I suggest that the research question and the hypotheses allied to the Theory used be presented more clearly. In certain passages the paragraphs are quite long which can interfere with the fluidity of the reading.

6. PLOS authors have the option to publish the peer review history of their article (what does this mean?). If published, this will include your full peer review and any attached files.

Reviewer #1: No

Reviewer #2: No

---

## [Author Response · Author response to Decision Letter 0]

23 Nov 2021

Please see attached 'author response' document

---

## [Decision Letter · Decision Letter 1]

13 Dec 2021

A socio-ecological examination of the primary school playground: primary school pupil and staff perceived barriers and facilitators to a physically active playground during break and lunch-times.

PONE-D-21-28206R1

Dear Dr. Graham,

We’re pleased to inform you that your manuscript has been judged scientifically suitable for publication and will be formally accepted for publication once it meets all outstanding technical requirements.

Kind regards,

Francisco Javier Huertas-Delgado, Ph.D.

Academic Editor

PLOS ONE

Additional Editor Comments (optional):

Reviewers' comments:

Reviewer's Responses to Questions

**Comments to the Author**

1. If the authors have adequately addressed your comments raised in a previous round of review and you feel that this manuscript is now acceptable for publication, you may indicate that here to bypass the “Comments to the Author” section, enter your conflict of interest statement in the “Confidential to Editor” section, and submit your "Accept" recommendation.

Reviewer #1: All comments have been addressed

Reviewer #2: All comments have been addressed

2. Is the manuscript technically sound, and do the data support the conclusions?

Reviewer #1: Yes

Reviewer #2: Yes

3. Has the statistical analysis been performed appropriately and rigorously? 

Reviewer #1: N/A

Reviewer #2: Yes

4. Have the authors made all data underlying the findings in their manuscript fully available?

Reviewer #1: Yes

Reviewer #2: Yes

5. Is the manuscript presented in an intelligible fashion and written in standard English?

Reviewer #1: Yes

Reviewer #2: Yes

6. Review Comments to the Author

Reviewer #1: After adjustments, I recommend that the editor consider the manuscript for publication in Plos One. I congratulate the authors for their work and for their responses to the review letter.

Reviewer #2: The authors adhered to my initial comments, which I think is important for the quality of the document. Refusing the theoretical approach used only changed the text.

7. PLOS authors have the option to publish the peer review history of their article (what does this mean?). If published, this will include your full peer review and any attached files.

Reviewer #1: No

Reviewer #2: No

---

## [Editor Report · Acceptance letter]

24 Jan 2022

PONE-D-21-28206R1 

A socio-ecological examination of the primary school playground: primary school pupil and staff perceived barriers and facilitators to a physically active playground during break and lunch-times. 

Dear Dr. Graham:

I'm pleased to inform you that your manuscript has been deemed suitable for publication in PLOS ONE. Congratulations! Your manuscript is now with our production department. 

Kind regards, 

on behalf of

Dr. Francisco Javier Huertas-Delgado 

Academic Editor

PLOS ONE